# First Study on *Gyrodactylus* (Monogenea: Gyrodactylidae) in Morocco, with Description of a New Species from *Luciobarbus pallaryi* and *Luciobarbus ksibi* (Actinopterygii: Cyprinidae)

**DOI:** 10.3390/ani13101624

**Published:** 2023-05-12

**Authors:** Miriam Isoyi Shigoley, Imane Rahmouni, Halima Louizi, Antoine Pariselle, Maarten P. M. Vanhove

**Affiliations:** 1Laboratory of Biodiversity, Ecology and Genome, Research Center Plant and Microbial Biotechnology, Biodiversity and Environment, Mohammed V University in Rabat, Rabat 10000, Morocco; 2Research Group Zoology, Biodiversity & Toxicology, Centre for Environmental Sciences, Hasselt University, Agoralaan Gebouw D, 3590 Diepenbeek, Belgium; 3Department of Veterinary Management of Animal Resources, Faculty of Veterinary Medicine, Liège University, 4000 Liège, Belgium; 4ISEM, Université de Montpellier, CNRS, IRD, 34095 Montpellier, France

**Keywords:** Cyprinidae, ectoparasite, Gyrodactylidea, *Luciobarbus*, Monopisthocotylea, North Africa, parasite, Platyhelminthes, Maghreb

## Abstract

**Simple Summary:**

Monogenean flatworms are mainly parasitic in lower aquatic vertebrates including fish, anurans and chelonians. *Gyrodactylus* is one of the 23 genera of Gyrodactylidae. With only 41 species described, the total number of *Gyrodactylus* species described from African freshwater fish still remains low. The known species represent only a fraction of the expected species richness of *Gyrodactylus* in Africa. In this study, we examined the gills of 738 cyprinid specimens. We isolated 26 individuals belonging to *Gyrodactylus* from these hosts. Twelve of these from two host species were morphologically characterized and proposed to belong to one single newly described species. In view of the importance of the cyprinid–monogenean system in studying the aquatic biodiversity and biogeography of North Africa, the present study is a substantial contribution to the parasite species inventory of these fishes.

**Abstract:**

To date, 41 species of *Gyrodactylus* have been described from Africa. However, none of these have been reported in Morocco. After identifying and examining 738 cyprinid host specimens, 26 specimens belonging to *Gyrodactylus* were found to parasitize the gills of nine species of *Luciobarbus*, *Carasobarbus*, and *Pterocapoeta*. The current study provides new information about the presence of a new parasitic species in Morocco, the first to be characterized on a species level in the Maghreb region. It describes in detail 12 specimens of *Gyrodactylus* isolated from the gills of *Luciobarbus pallaryi* (Pellegrin, 1919) and *Luciobarbus ksibi* (Boulenger, 1905). Based on morphoanatomical observations, the characterization of the specimens collected indicates a species of *Gyrodactylus* that is new to science, described here as *Gyrodactylus nyingiae* n. sp. The new species is different from previously described gyrodactylids infecting African cyprinid hosts because it has a longer hamulus total length, a longer hamulus root, a downward projecting toe of the marginal hook, and a trapezium-shaped ventral bar membrane with a slightly striated median portion and small rounded anterolateral processes. This study increases the total number of *Gyrodactylus* spp. found in African cyprinids to four.

## 1. Introduction

Fisheries and aquaculture are important sectors that make a significant contribution by creating job opportunities for approximately 59.51 million people. These sectors consist of capture fisheries and aquaculture, which, respectively, employ 39.0 million and 20.5 million individuals worldwide according to FAO [1]. Morocco is considered one of the top producers of fishery resources, occupying the 13th place after Chile [2]. In 2018, the national fisheries production totaled a volume of 1,371,683 tons for a turnover of 11,579,544 thousand dirhams [2]. In freshwater, culture-based fisheries which are projected to generate 13,000 tons of fish annually in Morocco, is the main source of fish protein. This production is based on the routine stocking of cultured organisms, mainly cyprinids, in lakes and reservoirs [3]. Biogeographers in the Maghreb region have often focused on ichthyofaunal studies because of its geographical position between the African and Eurasian plates. Primary freshwater fishes are a suitable subject for historical biogeography due to their limited dispersal that is strictly restricted to fluvial basins, showing less capacity for trans-watershed dispersal [4]. However, the freshwater fish fauna of North Africa shows low diversity, which could probably reflect a long period of isolation during the Cenozoic Era [4].

The high level of endemism of cyprinid fishes in Morocco (20 endemic species) with representatives from the genera *Carasobarbus* Karaman, 1971, *Luciobarbus* Heckel, 1843, *Labeobarbus* Rüppell, 1835, and *Pterocapoeta* Günther, 1902, noted by Rahmouni et al. [5] is linked to the geological and climatic history of the Mediterranean biome, which have led to the endemic status of many species (animal or plant) present in these zones [6,7]. The number of studies on freshwater fish parasites has increased globally due to the growing interest in developing fisheries and aquaculture as affordable sources of protein to sustain the rapidly growing human population, especially in some African communities [8]. For the management of this resource, thorough knowledge of the taxonomy, distribution, biology and ecology of parasites is of paramount importance [8,9].

The cyprinid host/parasite system is a good model for studying evolutionary phenomena and determining speciation mechanisms. Gyrodactylids have the broadest host range of any monogenean family (found on 19 bony fish orders), encompassing both narrowly specific and generalist species [10]. Due to their distinct mode of reproduction, they provide valuable insights into parasite speciation processes [10,11]. Despite there being various studies on *Gyrodactylus* von Nordmann, 1832, in Africa, and also on cyprinids, these often do not include species-level identifications (e.g., Allalgua et al.) [12] and only a few of its representatives have been identified on the species level in the whole continent. Currently, there are over 400 valid species of *Gyrodactylus* described [13]. In African freshwater fishes, only 41 species of *Gyrodactylus* have been described [14]. The known species represent only a fraction of the expected *Gyrodactylus* spp. in Africa [15]. Only three *Gyrodactylus* species have been described from small African cyprinids, always with a host belonging to *Enteromius* Cope, 1867; namely, *G. ivindoensis* Price and Gery, 1968, from *Enteromius* cf. *holotaenia* (Boulenger, 1904) in Gabon, *G. kyogae* Paperna, 1973, from *Enteromius neumayeri* (Fischer, 1884) and *Enteromius perince* (Rüppell, 1835) in Uganda and *G. paludinosus* Truter, Smit, Malherbe and Přikrylová, 2021 [14], from *Enteromius paludinosus* (Peters, 1852) in South Africa. In Morocco, no research has documented species of *Gyrodactylus* to date. Monogeneans belonging to *Gyrodactylus* are major pathogens in fishes as well as a major challenge in both fisheries and aquaculture. They are commonly found on the skin and fins of freshwater fishes but may be occasionally found on the gills [16,17,18]. Gyrodactylids have a level of economic significance that outweighs that of any other monogenean family. In Norway, for example, the introduction of *Gyrodactylus salaris* Malmberg, 1957, into the salmon industry resulted in uncontrollable epidemics and mortalities, leading to massive economic losses [19].

Despite the numerous economic benefits a country may achieve from the introduction of living organisms, they can also be detrimental to native species [20]. In Lake Naivasha, Kenya, for example, the common carp *Cyprinus carpio* Linnaeus, 1758, was thought to have reached the lake in 1999 during the heavy rains from juveniles that escaped in the Malewa River [21]. Parasitological studies on parasites of *C. carpio* in Lake Naivasha discovered that it is dominated by representatives of *Dactylogyrus* [22] with a high prevalence of 99.3% [22]. Moroccan irrigation channels and reservoirs have also been stocked with non-native freshwater fish species such as the silver carp *Hypophthalmichthys molitrix* (Valenciennes, 1844), common carp *Cyprinus carpio*, goldfish *Carassius auratus* (Linnaeus, 1758) and grass carp *Ctenopharyngodon idella* (Valenciennes, 1844). These fishes play a central role in aquatic ecosystems, especially with respect to their role as consumers in food chains. Their importance is increasingly recognized, making them a central focus in conservation, pollution prevention and restoration in aquatic ecosystems [23]. Despite the importance fishes offer, the introduced fishes could pose a threat to native fishes by providing a perfect opportunity for parasite transmission [24]. For this reason, it is important to have a baseline for the Moroccan native monogenean fauna of cyprinids. Therefore, and in view of the importance of the cyprinid–monogenean system in investigating the aquatic biodiversity and biogeography of North Africa, the present study aims to identify *Gyrodactylus* isolated from the gills of *Luciobarbus pallaryi* (Pellegrin, 1919) and *Luciobarbus ksibi* (Boulenger, 1905) in Morocco and contribute to the parasite species inventory of these fishes. 

## 2. Materials and Methods

### 2.1. Sample Collection

During September 2014 and June 2021, a total of 28 localities covering nine different watersheds in Morocco were sampled on five different occasions for cyprinid specimens, as shown in Figure 1. The fish specimens were collected after obtaining the required permit from the Ministry of Water, Forestry and Desertification Control (sampling permit no. 62 HCEFLCD/DLCDPN/CPC/PPC). These fish samples were collected using a backpack electrofisher (Samus-725G) or gill nets when the physicochemical water parameters could not allow sampling using the electrofisher. Fish hosts were identified morphologically following [25], euthanized by severing their spinal cords and dissected immediately. The gills were fixed in accordance with [26] and some fish specimens were frozen in a portable freezer and analyzed in the laboratory. The nomenclature and the classifications of fishes are those provided in [27]. The map showing sampling localities (Figure 1) was created using QGIS v3.22.8 (QGIS Development Team 2022, QGIS Information System, Open Source Geospatial Foundation Project. http://qgis.osgeo.org, accessed on 20 January 2023).

### 2.2. Parasitological Examination

The fish samples were transported to the laboratory for parasitological examination. Monogeneans were isolated under a dissecting microscope (Wild Heerbrugg) from the gills (gill arches from the right side of the excised fish). With the aid of a fine needle, the parasites were picked out one by one, subsequently mounted on a glass slide and then covered with a coverslip. The slides were mounted in accordance with [28]. For worms fixed in ethanol, Hoyer’s chloral hydrate was used [29] while ammonium picrate glycerine was used for frozen parasites [30]. The glass slide was left to dry for 24 h in a horizontal position before sealing the coverslip with Glyceel [31]. The type material was deposited in the collections of the research group Zoology: Biodiversity and Toxicology at Hasselt University (HU) (Diepenbeek, Belgium) (HU 838-841) and the Institut Scientifique of the Mohammed V University in Rabat (Rabat, Morocco) (ZA PPM 0101).

### 2.3. Identification of Representatives of *Gyrodactylus*

*Gyrodactylus* was distinguished from the other monogeneans as its members have a cylindrical body bearing two small cephalic lobes on the exterior part of the body, lack eyes and possess an opisthaptor armed with a single pair of hamuli linked by dorsal and ventral bars with 16 articulated marginal hooks (14 hooks in members of *Dactylogyrus*, the other monogenean genus most common on Moroccan cyprinids) [32].

### 2.4. Infection Parameters

Infection parameters, i.e., prevalence (P), mean intensity (M.I) and mean abundance (M.A) for members of *Gyrodactylus*, were calculated according to Bush et al. [33]. 

### 2.5. Morphological Characterization of Members of *Gyrodactylus*

Light microscopy using both phase and differential interference contrast approaches was used to study the shape and dimensions of sclerotized structures, which were viewed under a ×100 oil immersion objective on a Leica DM2500 optical microscope using Las X software v3.6.0.20104 fitted with a Leica DMC4500 camera. The whole mount, attachment organ, and male copulatory organ (MCO) (when present) on each specimen were photographed. The haptoral morphometrics (26 point-to-point measurements) followed the measurements proposed by [34]; these were taken using ImageJ v1.53k software (available at http://imagej.nih.gov/ij accessed on 15 September 2021) and only for specimens preserved using Hoyer’s solution. These measurements were given in micrometers (µm) as the mean, followed by the range in parentheses and the number of structures (*n*) measured for each metric. The micrographs taken were used to draw taxonomically important structures using Inkscape v1.2.

### 2.6. Statistical Analyses

For statistical analysis, a principal component analysis (PCA) was carried out in R Studio v4.1.0. The analyses included 19 measurements of the haptoral hard parts of hamuli and marginal hooks only. The MCO, ventral bar and dorsal bar measurements were excluded from the analysis due to the large number of missing data.

## 3. Results

### 3.1. Specimens Examined and Individuals of *Gyrodactylus* Isolated

A total of 738 fish specimens belonging to three genera (*Luciobarbus* Heckel, 1843, *Carasobarbus* Karaman, 1871, and *Pterocapoeta* Günther, 1902) were collected. Thirteen cyprinid fish species were identified and their gills were examined for infection with species of *Gyrodactylus* (Table 1). A total of nine out of the 13 species were found to be infected with representatives of *Gyrodactylus* (*n* = 26).

### 3.2. Infection Parameters

The infection parameters of examined hosts are shown in Table 2.

### 3.3. Characterization of a New Species of *Gyrodactylus*

All the isolated flatworms belonging to *Gyrodactylus* showed the diagnostic features of this genus: gyrodactylid monogeneans with an opisthaptor with one pair of haptoral anchors surrounded by 16 marginal hooks. The measurements are given in Table 3.

Class: Monogenea Van Beneden, 1858.

Subclass: Polyonchoinea Bychowsky, 1937.

Order: Gyrodactylidea Bychowsky, 1937.

Family: Gyrodactylidae Van Beneden and Hesse, 1863.

Subfamily: Gyrodactylinae Van Beneden and Hesse, 1863.

Genus: *Gyrodactylus* von Nordmann, 1832.

Species: *Gyrodactylus nyingiae* n. sp.

Type material: holotype (HU_838_IV.1.18) and five paratypes (HU_839_IV.1.19, HU_840_IV.1.20, HU_841_IV.1.21, ZA PPM 0101).

Type host: *Luciobarbus pallaryi* (Pellegrin, 1919) (teleostei: Cyprinidae).

Other host: *Luciobarbus ksibi* (Boulenger, 1905) (teleostei: Cyprinidae).

Type locality: Oued Guir (31°52′12″ N, 003°0′00″ W) (on type host).

Other locality: Oued Ksob (31°27′50.7″ N, 009°45′25.3″ W) (on *L. ksibi*).

Site of infection: Gill filament.

ZooBank registration: The Life Science Identifier (LSID) of the article is urn:lsid:zoobank.org:pub:15E78B1A-5DF7-4E37-935B-155A658FED77. The LSID for *Gyrodactylus nyingiae* Shigoley, Rahmouni, Louizi, Pariselle and Vanhove n. sp. is urn:lsid:zoobank.org:act:5E58B4AA-15F6-4540-BD49-559D023A56AA.

Studied material: 12 mounted individuals were measured; 11 of these were isolated from *L. pallaryi* and one was isolated from *L. ksibi.*

Etymology: The species epithet honors Dr. Dorothy Wanja Nyingi, an ichthyologist at the National Museums of Kenya and author of the first Guide to Common Freshwater Fishes of Kenya.

Description: Elongated body. A male copulatory organ (MCO) was observed in five specimens, was spherical (Figure 2a(i) and Figure 3ii), was positioned posteriorly to the pharynx and was armed with one principal spine and a single row of 5–6 smaller spines (Figure 2a(i) and Figure 3ii). Hamuli were slightly slender with a pointed tip with a superficial root (Figure 2a(ii) and Figure 3i,iv). The anterior end where dorsal the bar attaches on the hamulus was prominent, creating a notch between the root and dorsal bar attachment point. The dorsal bar was simple and flexible. The ventral bar had small rounded anterolateral processes with a trapezoid-like membrane having a slightly striated median portion (Figure 2a(iii),b). The marginal hook shaft was approximately perpendicular to the base of the marginal hook sickle (Figure 2a(iv),b(C) and Figure 3iii,v). The sickle point was slightly curved and perpendicular to the base with its tip in line with the distal end of the toe. Overlapping measurements (Table 3) and the similarity in the shape of the marginal hook sickle (Figure 4) suggest that the worms infecting the two host species are conspecific.

#### Remarks

The comparison with other gyrodactylid species is based on the phenotypic similarities to known parasite species and their occurrence from related hosts. From the three species of *Gyrodactylus* recorded from cyprinids in Africa, the newly described species of *Gyrodactylus* can be differentiated by the longer hamuli; *G. nyingiae* n. sp. 76.5 (65.9–88.2) compared to a hamulus total length in *G. ivindoensis* of 55 (52–58), 32.1 (23–33) in *G. kyogae* and 43.3 (35.1–51.5) in *G. paludinosus*. Like *G. paludinosus*, *G. kyogae* has an upward projecting toe, in contrast to that of *G. nyingiae* n. sp. whose toe points downwards. Additionally, the MCO of *G. nyingiae* n. sp. has one principal spine and five to six smaller spines arranged in a single row (Figure 2 and Figure 3), in contrast to *G. kyogae*, which has an unarmed MCO [35]. *Gyrodactylus kyogae*, in contrast to the other three species, lacks a ventral bar membrane. *Gyrodactylus ivindoensis* has shorter marginal hooks and a total marginal hook length of 22 (21–24) compared to that of *G. nyingiae* n. sp., which is 34.8 (31.7–42.1). When comparing the relative length of the root to the hamulus total length respectively, *G. nyingiae* n. sp. (26.7 vs. 76.4), *G. ivindoensis* (19.4 vs. 55) and *G. paludinosus* (15.4 vs. 43.3) have similar ratios of the root length to the total hamulus length (ca. 1:2.8). *Gyrodactylus kyogae* (9.2 vs. 33.1) on the other hand has a different ratio of the relative root length to the total hamulus length (1:3.5).

Due to the important biogeographical connections between the Middle East and the Maghreb region during the Cenozoic period in the dispersal of freshwater fish fauna, it is interesting to compare the *Gyrodactylus* fauna of the Iranian region with the North African ones [4,36]. The freshwater species of *Gyrodactylus* mentioned by [37] and [38] were either known to be from Europe or Central Asia, or undescribed. It is therefore productive to compare *G. nyingiae* n. sp. with widespread Palearctic species of *Gyrodactylus*-infecting cyprinids, several of which are reminiscent of *G. nyingiae* n. sp. in marginal hook morphology. This includes *Gyrodactylus mutabilitas* Bychowsky, 1957, and *Gyrodactylus scardiniensis* Glaeser, 1974, which can both be distinguished from *G. nyingiae* n. sp. by virtue of their shorter hamulus root (max. of 20 in *G. mutabilitas* and max. of 23 in *G. scardiniensis* versus min. of 24 in *G. nyingiae* n. sp.), and *Gyrodactylus schulmani* Ling, 1962, which has a hamulus of a total length of a max. of 44, shorter than the minimum of 66 of *G. nyingiae* n. sp. A *Gyrodactylus* species described from a fish species endemic to Iran is *Gyrodactylus jalalii* Vanhove, Boeger, Muterezi Bukinga, Volckaert, Huyse and Pariselle, 2012, a parasite of the cichlid host *Iranocichla hormuzensis* Coad, 1982. It can easily be distinguished from *G. nyingiae* n. sp. by its more pronounced ventral bar auricles and the sub rectangular ventral bar membrane, which contrast the properties of *G. nyingiae* n. sp. including its small rounded anterolateral processes and trapezium-shaped ventral bar membrane. Following [39], *Gyrodactylus molnari* Ergens, 1978, infecting *Cyprinus carpio* Linnaeus, 1758, in Iraq has a shorter hamulus length (55–65) compared to *G*. *nyingiae* n. sp. (65.9–88.2). Additionally, *G. molnari* has a longer dorsal bar (15–18), compared to *G. nyingiae* n. sp. (9.9–13.4), and an entirely different shape of the marginal hook sickle. Following [40,41], *Gyrodactylus sprostonae* Ling, 1962, was found on *Cyprinus carpio* in Iran. It has a longer dorsal bar (17.4–20) compared to *G. nyingiae* n. sp. (9.9–13.4) and shorter total hamulus length (48.47–54.23) compared to *G. nyingiae* n. sp. (65.9–88.2). Due to the fact that the goldfish (*Carassius auratus*) has been widely introduced in many countries including Morocco, it is also interesting to compare *G. nyingiae* n. sp. with *Gyrodactylus kobayashii* Hukuda, 1940, previously isolated from goldfish in central China [42]. *G. nyingiae* n. sp. has a MCO which resembles that of *G. kobayashii* with both possessing one principal spine and 5–6 smaller spines. Both species also have slightly curved marginal hook sickles. However, the marginal hook sickle in *G. kobayashii* has a tip that terminates beyond the limits of its toe while that of *G. nyingiae* n. sp. has its tip in line with the distal end of the toe. Additionally, *G. nyingiae* n. sp. has a longer hamulus (76.4 vs. 59.3), a longer sickle (47.4 vs. 40.5), a longer hamulus root (26.7 vs. 21.6), and a longer marginal hook shaft (28.7 vs. 23.3) than *G. kobayashii* has. Therefore, *G. nyingiae* n. sp. can be distinguished from the aforementioned *Gyrodactylus* species by virtue of its longer total hamulus length, longer hamulus root, small rounded anterolateral process and trapezium-shaped ventral bar membrane.

### 3.4. Multivariate Statistics

The morphological variation of the 12 specimens of *Gyrodactylus* was visualized based on a PCA performed on 19 standardized haptoral morphometric characters. The first two principal component axes contributed to 25.9% and 19.6% of the variation, respectively (Figure 5).

The biplot shows no clear separation which includes all the 12 specimens belonging to *Gyrodactylus*. From the PCA biplot, we can confirm that we are dealing with a single species described herein as *Gyrodactylus nyingiae* n. sp. To better illustrate how the newly identified *Gyrodactylus nyingiae* n. sp. compares to other previously described species of *Gyrodactylus*, we performed a PCA analysis based on the mean values of 11 haptoral morphometric parameters, i.e., HPL, HSL, HRL, HTL, MHTL, MHSHL, MHSL, MHSPW, MHSDW, MHAD and VBTL (Table 4). 

The first two principal components explained 29.7% and 23.0% of the total variation, respectively (Figure 6). 

With the average values from the supplementary individuals, the results of the PCA indicate a distinct difference between the *Gyrodactylus* species that were previously described and the newly described species mentioned in this study.

## 4. Discussion

*Gyrodactylus nyingiae* n. sp. is the first described species of *Gyrodactylus* in Morocco and to the best of our knowledge is the first member of this genus to be identified on a species level in the Maghreb region. Monogenean parasites have been reported in Tunisian aquatic vertebrates, including both marine and freshwater hosts. However, not even a single *Gyrodactylus* species has been recorded from the examined hosts [43]. The species reported in the current study is also the first gyrodactylid to be described from *Luciobarbus* in Africa, as previous studies on gyrodactylids infecting cyprinids in Africa have focused on the small barbs belonging to *Enteromius*, with only three *Gyrodactylus* species having being described so far [14]. Since *Gyrodactylus* is a genus of monogeneans with high diversity and minimal morphological variation, it has become a common practice to use a combination of morphological and molecular information for describing and delimiting new species [44,45]. However, due to the limited number of specimens in our study, we opted to use all of the individuals for phenotypic characterization. This decision was based on the fact that our study aimed to unravel the diversity of branchial monogeneans in the hosts, which had not been previously documented.

The current study’s low number of gyrodactylids isolated from cyprinid hosts could be due to the fact that only the gills were examined for parasites. Similarly, Louizi et al. [46] found a species depauperate fauna and low abundances of gill-infecting monogeneans on native cichlid fishes in Morocco. In addition to the low prevalence and possible seasonality of members of *Gyrodactylus*, less research, a lack of reports on infections, a lack of understanding of relationships between these monogeneans and cyprinid hosts [14], and environmental conditions in Morocco’s freshwater ecosystems might limit the species richness and abundance of certain monogenean taxa. On the other hand, *Dactylogyrus* reaches higher species richness and higher infection intensities in Moroccan cyprinid–monogenean systems [5,47]. It is also worth noting that the low sample size of the present study is only an indicative value of the population size of the new species of *Gyrodactylus* in the host. 

More research is needed on the African continent to understand the relationship, evolutionary history, and development of gyrodactylids and their hosts, as it is endowed with a diverse endemic fish fauna that undoubtedly possess undiscovered parasite diversity [48,49].

## 5. Conclusion

Based on the morphoanatomical observation of opisthaptoral parts of 12 individuals of *Gyrodactylus* in the current study, we describe a new species infecting two cyprinid hosts for the first time in Morocco and the Maghreb region. The new species is different from previously described gyrodactylids infecting cyprinid hosts because it has a longer total hamulus length, a longer hamulus root, a downward projecting toe, trapezium-shaped ventral bar membrane with slightly striated median portion and small rounded anterolateral processes.

## Figures and Tables

**Figure 1 animals-13-01624-f001:**
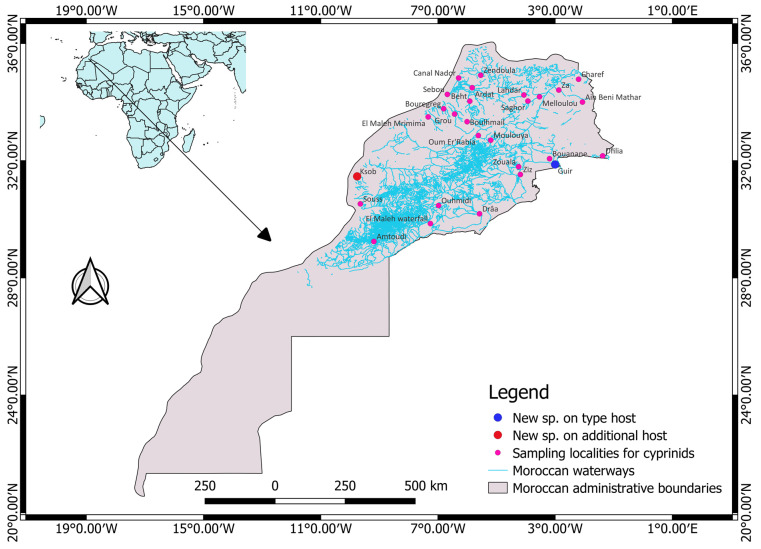
An overview of the sampling locations of cyprinid hosts examined for the presence of members of *Gyrodactylus*. The type host and additional host localities for the newly described *Gyrodactylus* species in the present study are depicted by blue and red circles, respectively.

**Figure 2 animals-13-01624-f002:**
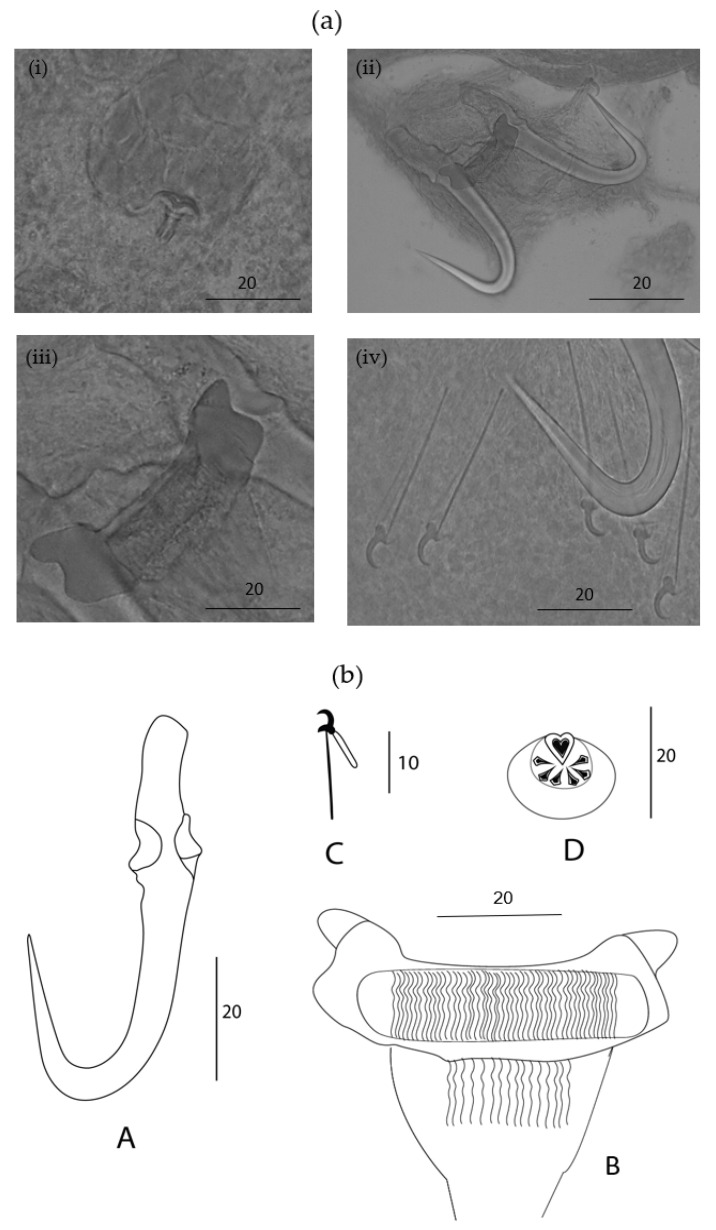
*Gyrodactylus nyingiae* n. sp. isolated from *Luciobarbus pallaryi*. (**a**) Micrograph of (**i**) male copulatory organ (MCO), (**ii**) hamuli, (**iii**) ventral bar, and (**iv**) marginal hooks (**b**) Drawings of sclerotized structures of the haptor with (**A**) hamulus, (**B**) ventral bar, (**C**) marginal hook, and (**D**) male copulatory organ. Scale bars are in µm.

**Figure 3 animals-13-01624-f003:**
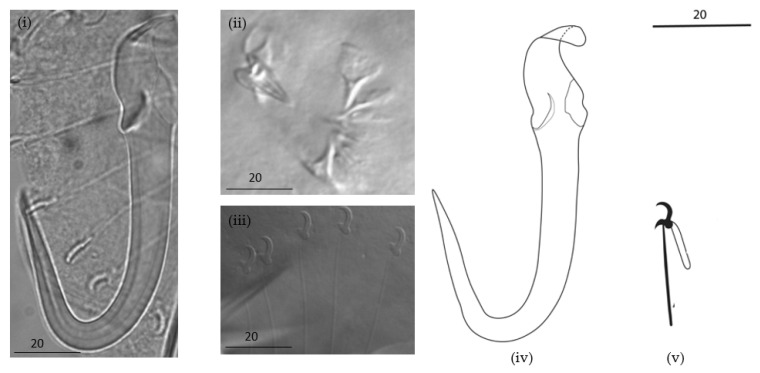
Micrograph and drawings of sclerotized structures (**i**,**iv**) hamuli, (**ii**) male copulatory organ, and (**iii**,**v**) marginal hooks of *Gyrodactylus nyingiae* n. sp. isolated from *Luciobarbus ksibi.* Scale bar represents 20 µm.

**Figure 4 animals-13-01624-f004:**
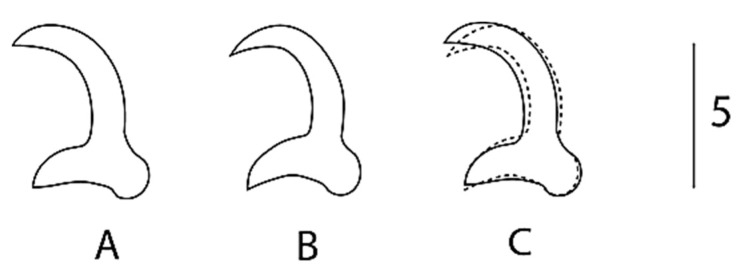
Overlay of marginal hook sickles (**C**) of *G. nyingiae* n. sp. from *L. ksibi* (**A**) and *L. pallaryi* (**B**) (dotted outline). Scale bar represents 5 µm.

**Figure 5 animals-13-01624-f005:**
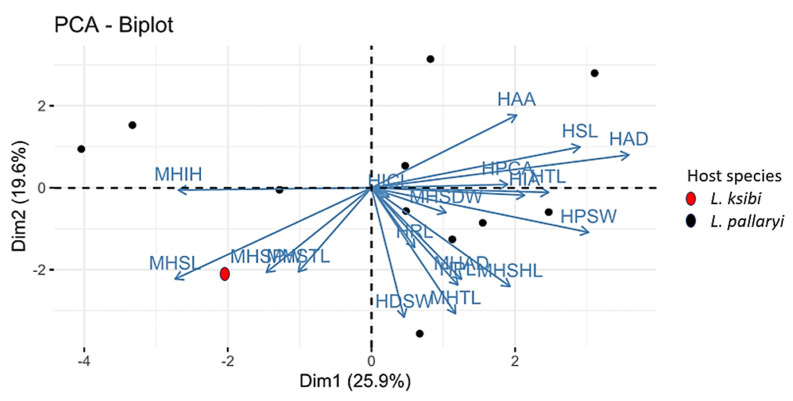
Biplot of PCA (first two axes) of all the 12 specimens of *Gyrodactylus nyingiae* n. sp. (For abbreviations of variables, please refer to Table 3).

**Figure 6 animals-13-01624-f006:**
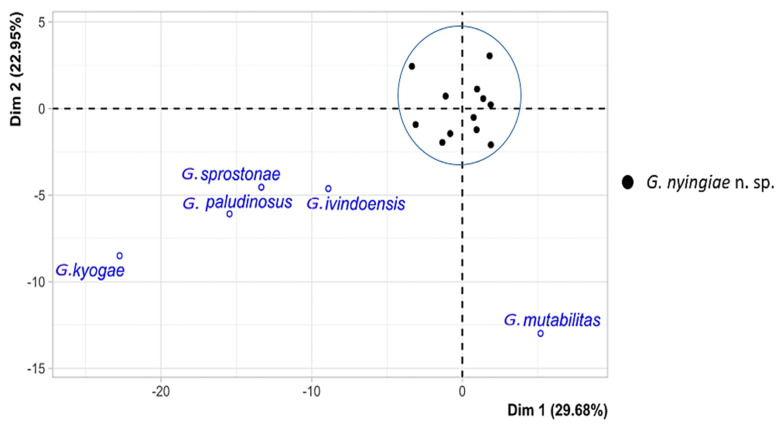
PCA (first two axes) of the newly described *Gyrodactylus* (in blue circle) and previously described species (supplementary individuals) based on average measurements of haptoral morphometric characters.

**Table 1 animals-13-01624-t001:** Cyprinid specimens collected, their localities and number of individuals belonging to *Gyrodactylus* infecting the hosts (note that the number of parasites collected might be an underestimation because only the right gill arches were screened for parasites).

			Coordinates			
Host	Locality	Watershed	Latitude	Longitude	No. of Hosts Sampled	No. of Specimens of *Gyrodactylus* Isolated from the Hosts
*Luciobarbus pallaryi* (Pellegrin, 1919)	Oued Guir	Ziz	31°52′12″ N	003°0′00″ W	157	14
	Oued Bouanane	Ziz	32°04′04″ N	003°11′23.9″ W		
	Oued Dfilia	Ziz	32°9′48.892″ N	001°22′37.4″ W		
*Luciobarbus rabatensis* Doadrio, Perea and Yahyaoui, 2015	Oued Grou	Bouregreg	33°35′28.0″ N	006°25′49.6″ W	24	3
	Oued Bouregreg	Bouregreg	33°46′18.0″ N	006°48′16.6″ W		
	Oued Boulhmail	Bouregreg	33°19′49.6″ N	006°00′15.1″ W		
*Luciobarbus maghrebensis* Doadrio, Perea and Yahyaoui, 2015	Oued Lahdar	Sebou	34°14′32.7″ N	004°03′53.9″ W	55	1
	Oued Saghor	Sebou	34°02′4.0″ N	003°55′45.5′W		
	Oued Ardat	Sebou	34°29′26.8″ N	005°49′49.2″ W		
	Oued Beht	Sebou	34°01′55.5″ N	005°54′43.2″ W		
	Oued Sebou	Sebou	34°15′48.0″ N	006°40′42.0″ W		
	Canal Nador	Sebou	34°49′19.7″ N	006°17′36.7″ W		
*Luciobarbus rifensis* Doadrio, Casal-López and Perea 2015	Oued Zendoula	Loukkos	34°54′57.6″ N	005°32′17.2″ W	19	3
*Luciobarbus guercifensis* Doadrio, Perea and Yahyaoui, 2016	Oued Melloulou	Moulouya	34°10′51.7″ N	003°31′59.6″ W	4	0
	Oued Za	Moulouya	34°24′38.9″ N	002°52′28.1″ W		
*Luciobarbus yahyaouii* Doadrio, Casal-López and Perea 2016	Oued Za	Moulouya	34°24′38.9″ N	002°52′28.1″ W	62	0
	Oued Charef	Moulouya	34°46′44.0″ N	002°11′56.0″ W		
	Oued Melloulou	Moulouya	34°10′51.7″ N	003°31′59.6″ W		
	Ain Beni Mathar	Moulouya	34°00′00.3″ N	002°03′58.6″ W		
*Luciobarbus zayanensis* Doadrio, Casal-López and Yahyaoui, 2016	Oued Oum Er’Rabia	Oum Er’Rabia	32°51′32.8″ N	005°37′18.9″ W	25	1
	Oued Moulouya	Moulouya	32°41′55.4″ N	005°11′51.2″ W		
*Luciobarbus lepineyi* (Pellegrin, 1939)	Oued Ziz	Ziz	31°31′34.7″ N	004°11′10.0″ W	127	0
	Oued Zouala	Ziz	31°47′31.9″ N	004°14′43.5″ W		
	Oued Dfilia	Ziz	32°9′48.892″ N	001°22′37.4″ W		
	Oued Drâa	Draa	30°11′12.24″ N	005°34′47.34″ W		
	Oued Ouhmidi	Draa	30°28′5.64″ N	006°58′36.12″ W		
	Oued El Maleh Mrimima	Draa	33°29’34.8’’N	007°19′58.1″ W		
	Oued El Maleh Waterfall	Draa	29°51′108″ N	007°15′23″ W		
	Oued Amtoudi	Draa	29°14′32.42″ N	009°11′8.71″ W		
*Carasobarbus moulouyensis* (Pellegrin, 1924)	Oued Moulouya	Moulouya	32°41′55.4″ N	005°11′51.2″ W	44	1
*Luciobarbus ksibi* (Boulenger, 1905)	Oued Oum Er’Rabia	Oum Er’Rabia	32°51′32.8″ N	005°37′18.9″ W	40	1
	Oued Ksob	Ksob	31°27′50.7″ N	009°45′25.3″ W		
*Luciobarbus massaensis* (Pellegrin, 1922)	Oued Souss	Souss-Massa	30°31′33.6″ N	009°38′53.6″ W	21	1
*Carasobarbus fritschii* (Günther, 1874)	Oued Grou	Bouregreg	33°35′28.0″ N	006°25′49.6″ W	157	0
	Oued Boulhmail	Bouregreg	33°19′49.6″ N	006°00′15.1″ W		
	Oued Lahdar	Sebou	34°14′32.7″ N	004°03′53.9″ W		
	Oued Oum Er’Rabia	Oum Er’Rabia	32°41′03.8″ N	005°13′00.3″ W		
	Oued Za	Moulouya	34°24′38.9″ N	002°52′28.1″ W		
	Oued Charef	Moulouya	34°46′44.0″ N	002°11′56.0″ W		
	Oued Ksob	Ksob	31°27′50.7″ N	009°45′25.3″ W		
	Oued Ardat	Sebou	34°29′26.8″ N	005°49′49.2″ W		
	Oued Beht	Sebou	34°01′55.5″ N	005°54′43.2″ W		
	Oued Sebou	Sebou	34°15′48.0″ N	006°40′42.0″ W		
*Pterocapoeta maroccana* Günther, 1902	Oued Oum Er’Rabia	Oum Er’Rabia	32°51′32.8″ N	005°37′18.9″ W	3	1
				Total	738	26

**Table 2 animals-13-01624-t002:** Prevalence, mean intensity and mean abundance of *Gyrodactylus* infecting the Moroccan cyprinids collected, based on examination of the right-side gill arches only (note that only localities where these parasites were found are retained in this table).

Locality	Species	H	N	*n*	P = (N/H) × 100	M.I = *n*/N	M.A = *n*/H
Oued Guir	*Luciobarbus* *pallaryi*	157	1	14	0.64	14	0.09
Oued Bouregreg	*Luciobarbus* *rabatensis*	24	1	3	4.17	3	0.13
Oued Sebou	*Luciobarbus* *maghrebensis*	55	1	1	1.82	1	0.02
Oued Zendoula	*Luciobarbus* *rifensis*	19	1	3	5.26	3	0.16
Oued Moulouya	*Luciobarbus* *zayanensis*	25	1	1	4.00	1	0.04
Oued Moulouya	*Carasobarbus* *moulouyensis*	44	1	1	2.27	1	0.02
Oued Ksob	*Luciobarbus ksibi*	40	1	1	2.50	1	0.03
Oued Souss	*Luciobarbus* *massaensis*	21	1	1	4.76	1	0.05
Oued Oum Er’Rabia	*Pterocapoeta* *maroccana*	3	1	1	33.33	1	0.33

H, number of examined hosts; N, number of infected hosts; *n*, number of individuals of *Gyrodactylus* in infected host; P, prevalence; M.I, mean infection intensity; M.A, mean abundance.

**Table 3 animals-13-01624-t003:** Morphometric measurements of sclerotized parts of *Gyrodactylus nyingiae* n. sp. The number of structures measured is given in superscript.

Host	*Luciobarbus pallaryi* (*n* = 1)	*Luciobarbus ksibi* (*n* = 1)	Both Host Species Combined
Total body length (TBL)	386.8 (278.3–456) ^5^	443.7	396.3 (278.3–456) ^6^
Total body width (TBW)	133 (115.8–145.9) ^6^	158	136.6 (115.8–158.4) ^7^
Hamulus total length (HTL)	76.5 (65.9–88.2) ^10^	75.3	76.4 (65.9–88.2) ^11^
Hamulus sickle length (HSL)	47.6 (42.5–54.8) ^8^	45.1	47.4 (42.5–54.8) ^9^
Hamulus aperture distance (HAD)	27.5 (21.1–30.2) ^9^	22.6	26.7 (21.1–30.2) ^10^
Hamulus point length (HPL)	36.9 (31.7–41.3) ^9^	36.3	36.2 (31.7–41.3) ^10^
Hamulus inner curve length (HICL)	1.7 (1.4–2.7) ^6^	4	2.1 (1.4–4) ^7^
Hamulus distal shaft width (HDSW)	5.5 (4.6–6.7) ^10^	6	5.7 (4.6–7.3) ^11^
Hamulus root length (HRL)	26.7 (24.2–28.3) ^7^	–	26.7 (24.2–28.3) ^7^
Hamulus aperture angle (HAA) (in degrees)	36.9 (31.5–45.4) ^7^	32.4	36.4 (31.5–45.4) ^8^
Hamulus point curve angle (HPCA) (in degrees)	4.4 (3.4–5.4) ^4^	–	4.4 (3.4–5.4) ^4^
Hamulus inner angle (HIA) (in degrees)	40.4 (36–45.4) ^7^	37.2	40 (36–45.4) ^8^
Hamulus proximal shaft width (HPSW)	10.4 (8.2–12.1) ^10^	10.3	10.2 (8.2–12.1) ^11^
Marginal hook total length (MHTL)	34.4 (31.7–42.1) ^8^	35.2	34.8 (31.7–42.1) ^9^
Marginal hook shaft length (MHSHL)	28.6 (26.1–33.4) ^9^	29.2	28.7 (26.1–33.4) ^10^
Marginal hook sickle length (MHSL)	6.2 (5.5–6.5) ^9^	6.6	6.3 (5.5–6.6) ^10^
Marginal hook sickle proximal width (MHSPW)	4.6 (3.9–5) ^9^	5.5	4.7 (3.9–5.5) ^10^
Marginal hook sickle distal width (MHSDW)	4.5 (3.9–5.1) ^9^	5	4.5 (3.9–5.1) ^10^
Marginal hook sickle toe length (MHSTL)	1.9 (1.8–2.1)^9^	2.1	2 (1.8–2.1) ^10^
Marginal hook aperture distance (MHAD)	5.5 (5–5.9) ^8^	5.3	5.4 (5–5.9) ^9^
Marginal hook in-step height (MHIH)	0.6 (0.5–0.9) ^8^	0.7	0.6 (0.5–0.9) ^9^
Ventral bar total length (VBTL)	19.6 (18.6–20.5) ^2^	-	19.6 (18.6–20.5) ^2^
Ventral bar total width (VBTW)	25.1 (24.8–25.4) ^2^	-	25.1 (24.8–25.4) ^2^
Ventral bar median length (VBML)	6.1 (5.5–6.8) ^3^	-	6.1 (5.5–6.8) ^3^
Ventral bar membrane length (VBMBL)	13.6 (12.7–14.5) ^3^	-	13.6 (12.7–14.5) ^3^
Ventral bar process length (VBPL)	3.7 (3.6–3.8) ^2^	-	3.7 (3.6–3.8) ^2^
Male copulatory organ diameter (MCO)	18.4 (16.5–19.5) ^4^	21.2	18.9 (16.5–21.2) ^5^
Principal spine length	6.5 (6.3–6.6) ^3^	6.5	6.5 (6.3–6.6) ^4^
Small spine length	3.3 (3.1–3.5) ^3^	5.4	4.4 (3.1–5.4) ^4^
Dorsal bar length (DBL)	11.9 (9.9–13.4) ^3^	-	11.9 (9.9–13.4) ^3^
Dorsal bar width (DBW)	1.6 (1.2–1.9) ^3^	-	1.6 (1.2–1.9) ^3^

**Table 4 animals-13-01624-t004:** Table showing the mean values of supplementary individuals included in the PCA.

	*G. paludinosus* Truter, Smit, Malherbe and Přikrylová, 2021	*G. kyogae* Paperna, 1973	*G. ivindoensis* Price & Gery, 1968	*G. sprostonae* Ling, 1962	*G. mutabilitas* Bychowskii, 1957
Reference	[14]	[14]	[14]	[40]	[38]
Country	South Africa	Uganda	Gabon	Iran	Iran
Host	*Enteromius paludinosus* (Peters, 1852)	*Enteromius neumayeri* (Fischer, 1884)	*Enteromius* cf. *holotaenia* (Boulenger, 1904)	*Cyprinus carpio* Linnaeus, 1758	*Vimba vimba* (Linnaeus, 1758)
HPL	19.1	14	21.4	22.8	23.8
HRL	15.4	9.2	19.4	16.6	-
HTL	43.3	33.1	55	52	67.5
HSL	33.8	28.2	37.6	39.8	25
MHTL	18.5	14.8	22	23.1	33.7
MHSHL	14.2	11.3	24.7	19.4	-
MHSL	4.4	3.1	5.5	4.5	10.3
MHSPW	2.5	2.4	3.3	3.2	-
MHSDW	1.9	2.1	2.5	3.1	-
MHAD	4.1	3.7	5.3	-	-
VBTL	17.4	4.8	18	19.7	28.5

## Data Availability

Type material was deposited in the collection of the research group Zoology: Biodiversity and Toxicology of Hasselt University (Diepenbeek, Belgium) (HU 838-841) and the Institut Scientifique of the Mohammed V University in Rabat (Rabat, Morocco) (ZA PPM 0101).

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
