# Peer review of "First Study on Gyrodactylus (Monogenea: Gyrodactylidae) in Morocco, with Description of a New Species from Luciobarbus pallaryi and Luciobarbus ksibi (Actinopterygii: Cyprinidae)"

_animals, 2023, doi:10.3390/ani13101624_

Round 1
Reviewer 1 Report
The manuscript provides new information about the presence of new parasitic species in Morocco.
I think that adding this type of information to the scientific literature is very interesting, however, the information presented can be clarified in some points.
The title seems very broad to me, since finally the new species of parasite is described in cyprinids of the genus Luciobarbus.Regarding the methodology, can the manipulations of the specimens modify the measurements that are made a posteriori or are they techniques widely used to apply morphometric techniques?
Regarding the methodology, can the manipulations of the specimens modify the measurements that are made a posteriori or are they techniques widely used to apply morphometric techniques?
In the section on infection parameters of materials and methods, the authors refer to citation 27 and table 2. This table is left far away in the manuscript. To facilitate reading, it could be briefly included in this section which are the measured parameters.
Table 1 is confusing, since it is not clear which information belongs to each line, nor to which coordinates the number of hosts sampled belongs.
In the remarks section, the authors indicate that they make a phenotypic comparison with other species. It would also be interesting if the multivariate analysis that they propose later were made by comparing the 12 specimens with species already described, to see how they are grouped in a cloud of points, and how they differ from the other species of the genus.
Have the authors considered the possibility of including molecular information and making comparisons with other species of the same genus? Today it is highly valued information.
Check the caption of figure 5 (says figure 1).
Author Response
Thank you for your comments on our manuscript. Please find attached responses to the comments you had on the paper.
Kind regards.

Reviewer 2 Report
Please provide details of the animal ethic approval for this study.
Line 62: for citation number 4, the name of the authors should be mentioned.
Line 65: affordable might be a better term than cheap, in this context?
Lines 68 to 70: This is true but is irrelevant to this manuscript and should be deleted.
Lines 106 to 111: the aim of the study is confusing. It has to be either “ to identify Gyrodactylus species infecting cyprinids across Morocco” or “to identify Gyrodactylus isolated from the gills of Luciobarbus pallaryi (Pellegrin, 1919) and Luciobarbus ksibi (Boulenger, 1905).” I think the latter is the right aim to keep. Please also add a brief statement about significance of these fish.
Lines 262 to 265: Authors mentioned it is relevant to compare the gyrodactylid fauna between Iran and North Africa but cited only two articles. They also should read and cite these ones:
Daghigh Roohi J, et al. (2019) Morphometric and Molecular identification of Gyrodactylus sprostonae in Guilan Province warm water fishes with an attitude of intensity and prevalence in selected farms. Iranian Veterinary Journal 15(2):50-58 doi:10.22055/ivj.2018.105207.1980
Daghigh Roohi J, et al (2019) Occurrence of dactylogyrid and gyrodactylid Monogenea on common carp, Cyprinus carpio, in the Southern Caspian Basin. Parasitology International:101977 doi:https://doi.org/10.1016/j.parint.2019.101977
Author Response
General comment 1
Please provide details of the animal ethic approval for this study.
Response: In response to your question, when procedures are carried out on live animals abroad, the Ethical Committee for Animal Experimentation of Hasselt University does not require ethical clearance.
General comment 2
Line 62: for citation number 4, the name of the authors should be mentioned.
Response: Thank you. This has been corrected and authors mentioned.
General comment 3
Line 65: affordable might be a better term than cheap, in this context?
Response: Thank you for the suggestion. This has been taken into consideration.
General comment 4
Lines 68 to 70: This is true but is irrelevant to this manuscript and should be deleted.
Response: Thank you. This has been deleted and a few additional lines to support the previous line added.
General comment 5
Lines 106 to 111: the aim of the study is confusing. It has to be either “ to identify Gyrodactylus species infecting cyprinids across Morocco” or “to identify Gyrodactylus isolated from the gills of Luciobarbus pallaryi (Pellegrin, 1919) and Luciobarbus ksibi (Boulenger, 1905).” I think the latter is the right aim to keep. Please also add a brief statement about significance of these fish.
Response: Thank you for the suggestion. We have adapted the latter suggestion in order to be more precise in our study objective.
General comment 6
Lines 262 to 265: Authors mentioned it is relevant to compare the gyrodactylid fauna between Iran and North Africa but cited only two articles. They also should read and cite these ones:
Daghigh Roohi J, et al. (2019) Morphometric and Molecular identification of Gyrodactylus sprostonae in Guilan Province warm water fishes with an attitude of intensity and prevalence in selected farms. Iranian Veterinary Journal 15(2):50-58 doi:10.22055/ivj.2018.105207.1980
Daghigh Roohi J, et al (2019) Occurrence of dactylogyrid and gyrodactylid Monogenea on common carp, Cyprinus carpio, in the Southern Caspian Basin. Parasitology International:101977 doi:https://doi.org/10.1016/j.parint.2019.101977
Response: Thank you for the articles recommendation. We read them and included the described gyrodactylid monogeneans in our manuscript in the remarks section line 321-323.
Reviewer 3 Report
First Study on Gyrodactylus (Monogenea: Gyrodactylidae) in Morocco, with Description of a New Species from Cyprinids (Actinopterygii: Cyprinidae)
AuthorsThis manuscript deals with description of a new species of monogenean genus Gyrodactylus from the gills of freshwater Cyprinid fishes of Morocco. Description and delimitation of species is presented by using morphological traits. Moreover, infection parameters, host relationships and spatial distribution data for the monogenean species are briefly noted. The morphological description is well done and complete, the figures and pothograps of the specimens are fine.
However, I have two main concerns regarding this manuscript.
1st Gyrodactylys is one of the most species rich genera within the Monogenea, however with a minimal morphological variability. Due to the anterior, the usage of a combination of morphological and molecular information have become a common practice in the description and delimitation of a new species (for example Razo-Mendivil et al., 2016. Parasitology International 65: 389-400 http://dx.doi.org/10.1016/j.parint.2016.05.009; Pinacho-Pinacho et al. 2021. Parasitology Research 120: 831-848. We strongly recommend the author to include molecular data (sequences of nuclear 18S rNA, 28S, ITS1, ITS2 as well as COI markers have been used to characterize species of this genus) to support to erect a new species.
2nd The monogeneans Gyrodactylus spp. are typically found on the skin and fins of its hosts (Buchmann and Bresciani, 1997. Parasitology Research 84: 17-24; Chen et al., 2019. Aquaculture 502: 176-188 http://doi.org/10.1016/j.aquaculture2018.12.018; Rubio-Godoy et al., 2012. Veterinary Parasitology 183: 305-316. Moreover, the authors of this Ms report they isolated monogeneans only from gill arches from the right side (lines 129-130). Both these facts explain the very low numbers of infection here reported by this claimed new species. We acknowledge the authors recognize that (lines 305-306), however, a more emphatic explanation of all these is in order; because the reporting of numbers (Table 1) and spatial distribution (Map) and text, could lead further to an erroneous understanding of this infection.
Minor points:
It is not clear from my point of view why the authors describe from PCA analysis and biplot, a single cloud of points including all their 12 specimens, and conclude that they were dealing with a single species. Furthermore, the 19 measurements of the haptoral hard parts and marginal hooks are not identified (biplot). The same number Fig. 1 is given to the map and the biplot.
Data in Table 2 make it clear that the specimens were collected from 9 locations, which do not coincide with the information given in the map (current Fig. 1). Perhaps it could be related to information given in text line 209, but it is confused as currently presented (when comparing map and table data). Furthermore, green and blue color points are not easily visible in the figure.
Table 1. Please change abbreviations Nb. to No. or Num. which are of common usage.
Author Response
Thank you for the comments on our manuscript. Please find the attached response to the paper.
Kind regards.

Round 2
Reviewer 2 Report
I am with the amendments.
Author Response
Thank you for accepting our amendments.
Kind regards,
Miriam.
Reviewer 3 Report
Accepted after minor revision.
Authors please take care of the next:
Line 99, please include “..Monogeneans of the genus Gyrodactylys are typically found on the skin and fins of mostly freshwater fishes (add the pertinent references please Buchmann and Bresciani, 1997. Parasitology Research 84: 17-24; Chen et al., 2019. Aquaculture 502: 176-188 http://doi.org/10.1016/j.aquaculture2018.12.018; Rubio-Godoy et al., 2012. Veterinary Parasitology 183: 305-316).
About line 104 please include some explanation as the following example: Due to the fact that Gyrodactylys is one of the most rich genera of monogenean and with a minimal morphological variability, the usage of a combination of morphological and molecular information have become a common practice in the description and delimitation of a new species of Gyrodactylys However, herewith we do not have enough specimens, or …… please authors give your reasons why do not include molecular data and argue your points of view for erecting and naming a new species relaying only in morphology and traditional taxonomy technics.
Line 148 delete “to flatten the specimens” please. It cast doubts about the value and exactitude of measurements.
Line 186. Add details about the supplementary individuals and specimens previously described (lines 346, lines 341-349), give the species names here, the measures used and the procedence of the data taken.
Lines 371 – 381 I would discuss again the speculations (however acceptable) regarding the lower numbers (prevalence and abundance) recorded for the new named species. The pointing out of possible seasonality, or the environmental conditions of the freshwater systems must be clearly confronted against the fact that sampling of gills are not the most adequate procedures to give a picture of the extension and distribution of such and infection by Gyrodactylus monogeneans; i.e. the authors must clearly acknowledge that its data are only indicative, they only can suggest low numbers. Others (me included) could interpret that such low numbers are indicative of a higher real infection by this species.
I acknowledge that the second PCA included (current Fig. 6) give a strong support to the proposal of the authors for erecting a new species. Good point.
Author Response
Please find attached document regarding the comments you gave earlier
Kind regards,
Miriam
